# Cotton Seedling Detection and Counting Based on UAV Multispectral Images and Deep Learning Methods

**Yingxiang Feng** [1,†], **Wei Chen** [1], **Yiru Ma** [1], **Ze Zhang** [1], **Pan Gao** [2,†] and **Xin Lv** [1,*]

[1] Xinjiang Production and Construction Crops Oasis Eco-Agriculture Key Laboratory, College of Agriculture, Shihezi University, Shihezi 832003, China; fengyingxiang@163.com (Y.F.); m17792619205@163.com (W.C.); mayiru@stu.shzu.edu.cn (Y.M.); zhangze1227@shzu.edu.cn (Z.Z.)

[2] College of Information Science and Technology, Shihezi University, Shihezi 832003, China; gp_inf@shzu.edu.cn

[*] Correspondence: luxin@shzu.edu.cn

[†] These authors contributed equally to this work.

**Abstract:** Cotton is one of the most important cash crops in Xinjiang, and timely seedling inspection and replenishment at the seedling stage are essential for cotton's late production management and yield formation. The background conditions of the cotton seedling stage are complex and variable, and deep learning methods are widely used to extract target objects from the complex background. Therefore, this study takes seedling cotton as the research object and uses three deep learning algorithms, YOLOv5, YOLOv7, and CenterNet, for cotton seedling detection and counting using images at six different times of the cotton seedling period based on multispectral images collected by UAVs to develop a model applicable to the whole cotton seedling period. The results showed that when tested with data collected at different times, YOLOv7 performed better overall in detection and counting, and the T4 dataset performed better in each test set. Precision, Recall, and F1-Score values with the best test results were 96.9%, 96.6%, and 96.7%, respectively, and the $R^2$, RMSE, and RRMSE indexes were 0.94, 3.83, and 2.72%, respectively. In conclusion, the UAV multispectral images acquired about 23 days after cotton sowing (T4) with the YOLOv7 algorithm achieved rapid and accurate seedling detection and counting throughout the cotton seedling stage.

**Keywords:** cotton; seedling stage; count; detection; multispectral images; deep learning

## 1. Introduction

Cotton is the second largest crop after grain [1], and cotton production significantly impacts national economic development. Timely seedling inspection and replenishment during the cotton seedling stage is a prerequisite for ensuring high yield, quality, and efficiency in the later stages of cotton production [2]. Accurate estimates of the number of plants in a field crop is essential to predict yield [3,4]. Therefore, rapid and accurate detection and counting during the cotton seedling stage is essential. At present, the traditional counting method is mainly based on manual counting. Challenges such as high cost, high workload, poor timeliness, inaccurate counting, and poor data representativeness are not conducive to timely replanting. In contrast, UAV remote sensing-assisted counting can make up for the shortcomings of manual counting.

UAV remote sensing technology has recently been widely used in agriculture because of its low cost, high resolution, and fast and repeatable capture capability [5,6]. Compared with ground-based crewless vehicles and high-altitude satellite remote sensing images, UAVs are more suitable for farmland monitoring and are less affected by the atmosphere. Currently, there are more studies on monitoring seedling emergence and plant density of maize [7,8], cotton [9,10], and rice [11,12] based on UAV images. Thus, acquiring high-resolution images based on UAV remote sensing platforms is becoming an essential tool for crop emergence detection, crop growth monitoring, and yield prediction [13–15]. Most previous studies were based on RGB images acquired by UAVs for counting plants and seedlings [16–18].

However, when performing crop seedling counting, the high proportion of soil in the images and the strong influence of soil reflectance on crop color due to the small size of plants and the sensitivity of visible images to light changes pose a challenge for seedling detection and counting [19]. Furthermore, in the single-minded pursuit of high precision, the imaging requirements of RGB cameras are getting higher and higher; however, this pursuit also means an increase in cost. In contrast, the near-infrared (NIR) channel has a high reflectance and generally absorbs 10% or less of the radiation [20]. Compared with RGB images under low-illumination conditions, NIR images have a clear structure and no noise impairment [21,22]. Therefore, multispectral images with near-infrared (NIR) bands can be used more effectively for crop segmentation and counting.

The existing Phantom 4 Multispectral UAV imaging system, which does not require additional integration for image geotagging, has a low payload weight, is less burdensome to operate for acquisition [23], can also work under different lighting conditions, and is suitable for crop phenotyping studies [24]. Nevertheless, multispectral imaging systems are less used for plant seedling counting. The resolution of images acquired by RGB sensors at the same image size is lower than that of spectral sensors [25]. In contrast, hyperspectral images have a large amount of data and redundant information compared to multispectral images, which are more suitable for information extraction. Osco et al. [26] evaluated the performance of individual bands and their combinations using multispectral images in four bands (red, green, red-edge, and NIR) and the results showed that the combination of three bands, red, green, and NIR, had better performance. Multispectral imaging allows the calculation of multiple vegetation indexes, which can be used for plant phenotyping studies and the development of multivariate regression models [27]. In this multivariate approach, spectral, morphological, or textural information can be used individually or in combination with each other [28]. For example, random forests (RF) and support vector machines (SVM) are used to rapidly classify images [29]. Traditional machine learning algorithms such as random forest (RF) and support vector machine (SVM) can quickly classify images. Support vector machine models can be used to classify spectral and textural features and estimate seedling number and density [30]. For supervised classification methods, manually selecting a sufficient number of representative samples from many images is tedious and time-consuming, and the manual selection also reduces classification accuracy [31]. Therefore, more innovative algorithms are being created, tested, and improved; for example, based on deep learning models are used to build the model to identify methods with better generalizability.

Deep learning techniques can accurately extract target objects from complex backgrounds [32]. Compared to traditional methods, deep learning can automatically learn different features from images or datasets to identify and locate individual objects of interest [33]. Fan et al. [34] combined whole convolutional networks (FCNs) based on RGB images from drones to count beets, corn, and strawberries with an error of less than 4.6%. Barreto et al. [35] proposed a convolutional neural network (CNN)-based deep learning method that performs planting row identification and plant counting with high accuracy for maize and citrus. Experiments using deep learning algorithms to perform counting tasks have been conducted in various industries, but many challenges remain. For example, image resolution, weeds, plant size, plant density, and overlap directly affect detection and counting accuracies [36]. To improve detection accuracy, many researchers have developed and improved target detection algorithms according to their needs. YOLOv5, YOLOv7, and CenterNet are algorithms that have been applied to target detection tasks. YOLOv5 and YOLOv7 are anchor point-based target detection algorithms. These algorithms mainly generate candidate frames based on predefined anchor points, identify different sizes and aspect ratios of anchor points, compare them with genuine target frames, and consider them as target objects if they exceed a certain threshold; otherwise, they are considered as background or other objects. YOLOv5 has been applied to pest identification [37], tomato virus disease identification [38], and apple real-time target detection [39]. YOLOv7 has been applied to the estimation of the sparrow duck population [40], real-time detection of kiwi [41], and localization of road cracks [42]. CenterNet is an improved version of CornerNet, a key

point-based target detection algorithm. After the previous key point target detection algorithm, an additional essential point matching process, the NMS post-processing process is needed. In contrast, this algorithm only predicts the location of the center point of the bounding box. It uses the center point as the key point, eliminating the post-processing process and improving the detection speed. CenterNet has been applied to construction vehicle detection [43], cotton seedling counting [44], and wheat broadleaf weed seedling detection [45]. All three algorithms have great potential for plant detection and counting.

Therefore, the multispectral images acquired by the UAV at different times were used to construct seedling detection and counting models by (1) selecting the optimal algorithm using the performance of YOLOv5, YOLOv7, and CenterNet in the six time points of collected UAV multispectral images; (2) selecting the optimal cotton seedling detection and counting time based on the selected optimal algorithm; and (3) final selection of the cotton seedling detection and counting model applicable to the whole seedling stage. Altogether, the aim of this study was to identify a method to provide a fast, highly time-efficient decision and basis for later agricultural production management.

## 2. Materials and Methods

### 2.1. Experimental Sites

As shown in Figure 1, the experiment was conducted in April–May 2022 at the Second Company of Shihezi University Agricultural Experimental Farm (44°19′N, 85°59′E; 443 m above sea level). The varieties planted in the trial area were "Xinlu Early 53" and "Xinlu Early 84", and the planting pattern was "one film, three tubes, and six rows" with a comprehensive and narrow row design. The sowing date was 15 April 2022, and the seedling emergence rate was about 85% on April 25. The trial was conducted at the cotton's 2–4-leaf stage (including cotyledons). The specific sampling dates were: 25 April 2022 (T1, 10 days after sowing), 1 May 2022 (T2, 17 days after sowing), 5 May 2022 (T3, 21 days after sowing), 7 May 2022 (T4, 23 days after sowing), 9 May 2022 (T5, 25 days after sowing), and 12 May 2022 (T6, 28 days after sowing).

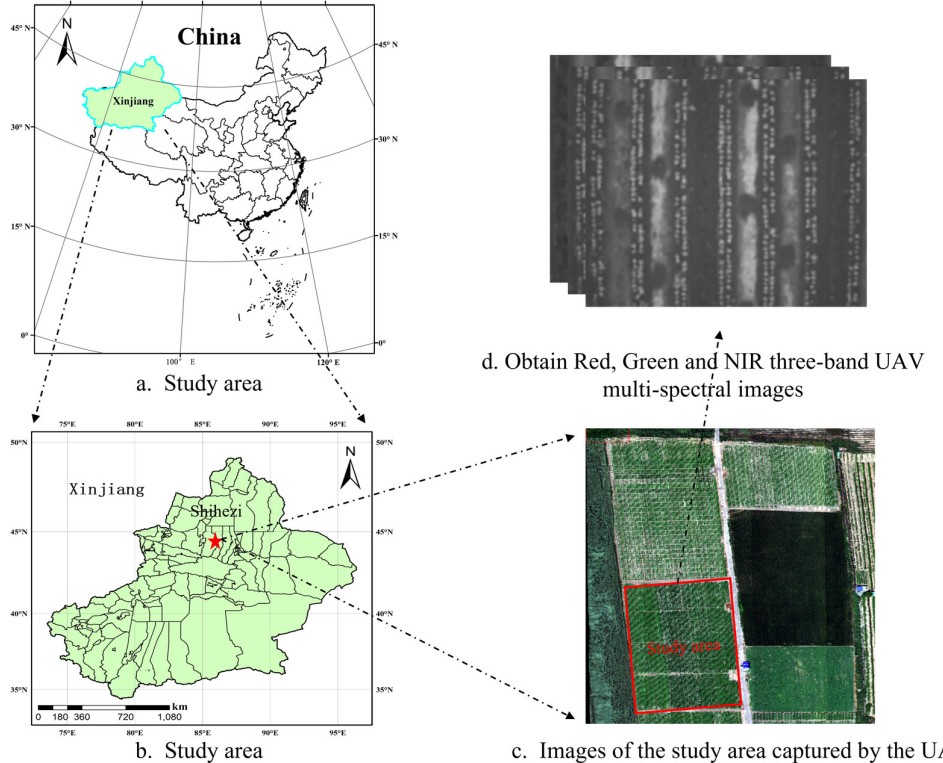

**Figure 1.** Study area overview: (**a**) is the map of Xinjiang, (**b**) is the map of Shihezi, (**c**) is the test site, and (**d**) is the multispectral images collected by the UAV.

### 2.2. Data Collection

2.2.1. Ground Data Acquisition

Company 2 of the Agricultural Experimental Farm of Shihezi University was the experimental site. The varieties planted in the experimental area were "Xinlu Early 53" and "Xinlu Early 84" from April to May 2022.

2.2.2. UAV Image Acquisition

The drone data acquisition was performed on the same day as the ground data acquisition. The lens was positioned vertically downward during image acquisition, and the interval was 2 s. The flight height was 10 m, the course overlap degree was 60%, and the side overlap degree was 60%. The parameters of the UAV are shown in Table 1. The camera parameters were as follows: FOV was 62.7°, the focal length was 5.74 mm, the aperture was f/2.2, and the monochrome sensors were 2 megapixels. The image resolution was 1600 × 1300 pixels, and the format was TIFF (multispectral imaging). Images of the red, green, and near-infrared bands were acquired, and the acquired images were stitched into orthophoto images using DJI Wisdom (China, DJI) software. The orthophoto images after stitching were stored in TIFF format.

**Table 1.** Phantom 4 Multispectral parameters.

| Parameter | Value |
|---|---|
| UAV weight | 1487 g |
| Max flight time | 27 min |
| Image max resolution | 1600 × 1300 (4:3.25) |

### 2.3. Image Processing

As shown in Figure 2, in the first step, the acquired UAV images were stitched into ortho images using DJI Smart Image (DJI, China) software, and the stitched orthoimages were stored in TIFF format. In the second step, the acquired multispectral images in the red, green, and near-infrared bands were band synthesized using ENVI5.3. In the third step, a cropping operation was performed on the pre-processed images using Python 3.8. This cropping operation removed the edge data with more distortion and anomalies during the stitching process, reducing the impact of related algorithm design and model construction.

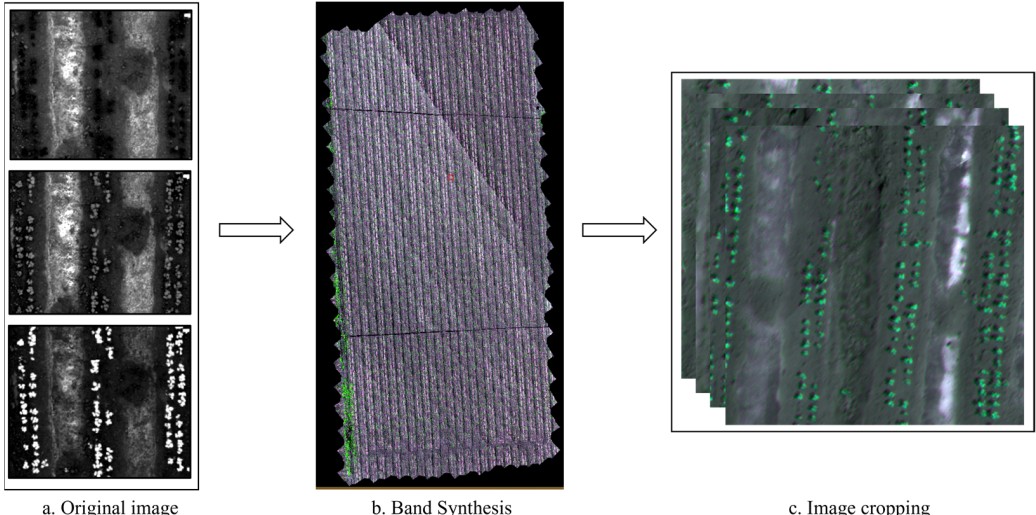

a. Original image    b. Band Synthesis    c. Image cropping

**Figure 2.** Image processing flow chart: (**a**) is the original images of the red, green and NIR bands, (**b**) is the image after three-band synthesis, (**c**) after pre-processing and cropping of the image to 640 × 640 size.

### 2.4. Dataset Creation

The segmented images were opened for labeling using the labelimg tool; 40 images were labeled each time, 240 images were labeled 6 times, and 37,840 cotton seedlings were labeled. After the labelling was completed, the six datasets were randomly divided according to the ratio of the training set/validation set = 3:1. We applied the validation sets from other time points as the test dataset at a particular time when performing the test. However, we tested the models individually each time instead of testing the models by combining the validation sets from several other times together, as shown in Table 2. For example, the test results for T1 were from testing the detection and counting models constructed for the T1 dataset against the validation sets for the T2, T3, T4, T5, and T6 datasets, individually.

**Table 2.** The number of cotton seedling marks in training, verification, and testing images at different time points.

| Time | Train Set | Val (Test) Set | Total |
|---|---|---|---|
| T1 | 3234 | 1143 | 4377 |
| T2 | 3715 | 1235 | 4950 |
| T3 | 6284 | 2108 | 8392 |
| T4 | 4192 | 1305 | 5497 |
| T5 | 4160 | 1408 | 5568 |
| T6 | 6770 | 2286 | 9056 |
| Total | 28,355 | 9485 | 37,840 |

T1–T6 were 10 days after seeding, 17 days after seeding, 21 days after seeding, 23 days after seeding, 25 days after seeding, and 28 days after seeding, respectively. Train, Val, Test, and Total represent the number of markers of cotton shoots in the training set, verification set, test set, and in total, respectively.

### 2.5. Deep Learning Models

#### 2.5.1. YOLOv5

As the basic version of the YOLO series, YOOv1 is a one-stage object detection algorithm that directly generates a location and category from pictures without a region proposal process. Compared with YOLOv1, YOLOv2 improves the detection performance. A common training method is proposed to train the target detector on detection and classification data. The network structure of YOLOv3 is divided into four parts [46]: Input Terminal, Backbone, Neck, and Prediction. Yolov4 has many innovations based on YOLOv3. For example, mosaic data are used for input; CSPDarknet53, Mish activation function, and Dropblock are used for the Backbone; SPP and FPN+PAN are used for Neck; and CIOU_Loss and DIOU_nms operations are used for the output. Compared with YOLOv4 [47], the input side of YOLOv5 is still mosaic data enhancement, while adding an adaptive anchor frame calculation, adaptive picture scaling, and other operations. The Backbone adopts a Focus structure and CSP structure. The Neck is still an FPN+PAN structure [48]. The Prediction part is also the GIOU_Loss operation. The network structure diagram is shown in Figure 3. The image input size is $640 \times 640$, the initial training weight file was yolov5s, the optimizer selected is SGD, the batch size is 16, and the epoch is 300.

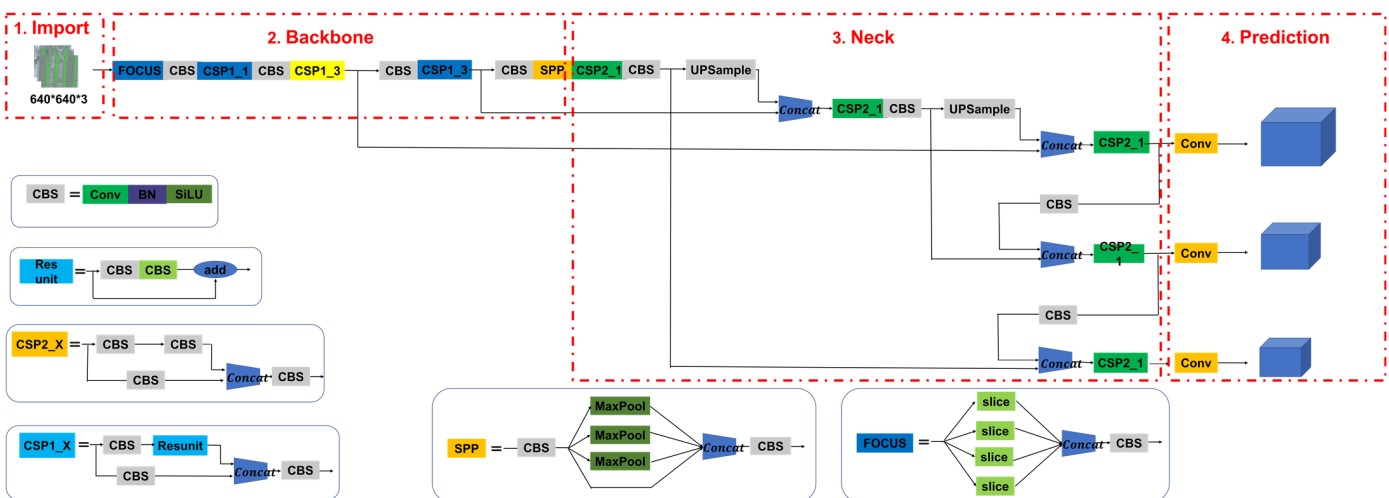

**Figure 3.** YOLOv5 network structure.

### 2.5.2. YOLOv7

YOLOv7 is inherited mainly from YOLOv5, including the overall network architecture, configuration file setup, training, reasoning, and verification processes. In addition, YOLOv7 also inherits a lot from YOLOR, including the design of different networks, the setting of hyperparameters, and the inclusion of implicit knowledge learning. It also imitates YOLOX's SimOTA strategy when matching positive samples. In addition to these structures in the existing YOLO version, YOLOv7 includes the latest tricks in recent years: efficient aggregation networks, reparameterized convolution, auxiliary training modules, model scaling, and more. In the efficient aggregation network the main factors to consider are the number of parameters, computation, and computation density. By controlling the shortest and longest gradient paths, deeper networks can learn efficiently and converge better. The role of reparameterization is to accelerate the network under the condition of ensuring the performance of the model. It mainly fuses the convolution +BN layer and different convolutions into a convolution module. The auxiliary training module uses deep supervision which is commonly used to train deep networks. The central concept is to add additional auxiliary heads in the middle layer of the network and shallow network weights guided by auxiliary losses. The role of model scaling is to scale the baseline up or down to make it applicable to different computing devices [49]. Model scaling methods usually include different scaling factors such as input size, depth, width, and stage, thus achieving a good trade-off in terms of the number of parameters, computation, reasoning speed, and network precision. The network structure diagram is shown in Figure 4. The image input size was 640 × 640, the initial training weights file was yolov7, the optimizer was adam, the batch size was 8, and the epoch was 1000.

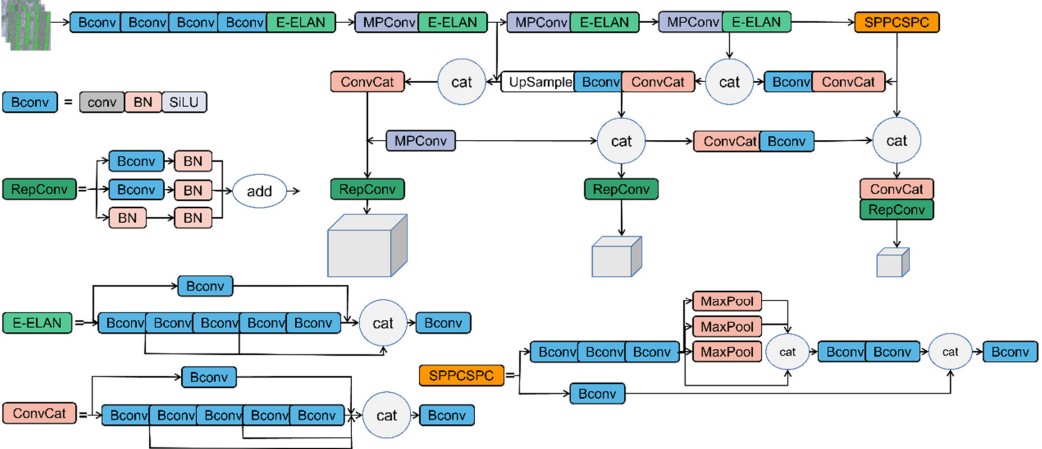

**Figure 4.** YOLOv7 network structure.

### 2.5.3. CenterNet

CenterNet is an anchorless target detection algorithm that is an improvement of the CornerNet algorithm. The algorithm removes inefficient and complex anchors, performs filtering operations directly on the heat map, removes time-consuming NMS post-processing operations, transforms the target detection problem into a critical point detection problem, and further improves the performance of the detection algorithm. In contrast to CenterNet, most target detection and recognition algorithms usually frame the target in a rectangular box. The horizontal and vertical axes of the rectangular box are parallel to the horizontal and vertical axes of the image. These detectors identify potential target locations and then classify them, which wastes time and requires additional post-processing. CenterNet, on the other hand, does not require manual thresholding of foreground or background by assigning anchors based on location rather than dense coverage on the feature map. Each object in it has only one positive sample of anchor, so there is no need for non-maximal value suppression processing, i.e., no post-processing methods are required, and only local peaks in the critical point heatmap need to be extracted. A larger output resolution is used, which eliminates the need for multiple anchors [50]. The primary process of building the target detection and recognition model is as follows: the image is fed into a fully convolutional network used to generate the heatmap, the peak position of the heatmap is the center of the target, and the image features at each peak position are used to predict the width and height of the target bounding box. Its network structure diagram is shown in Figure 5. The image input size was 640 × 640, the initial training weights file was centernet_resnet50_voc, the optimizer was chosen as adam, the batch size was 4, and the epoch was 900.

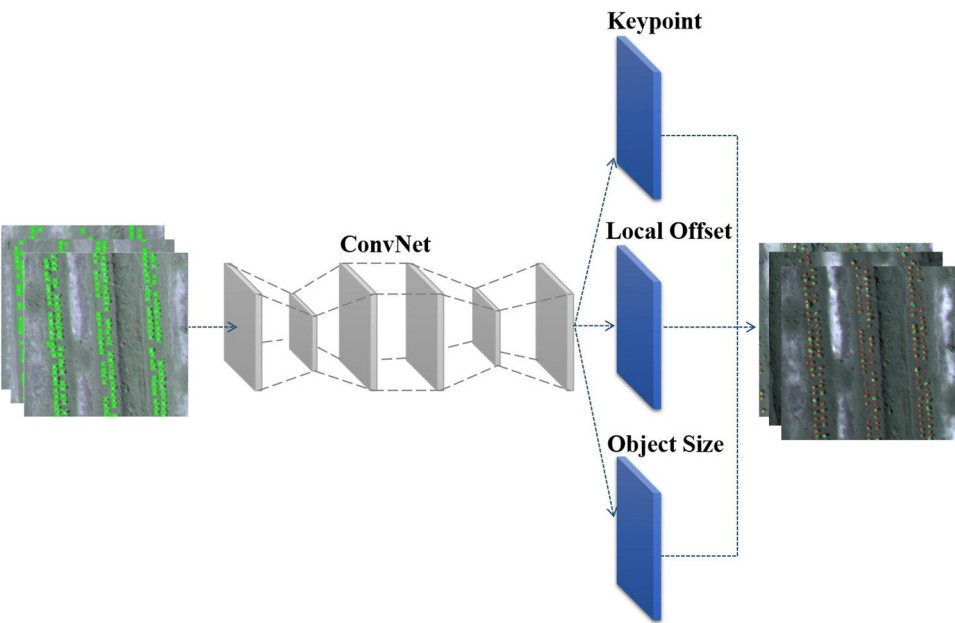

**Figure 5.** CenterNet network structure.

### 2.6. Precision Validation

In this study, *Precision*, *Recall*, and *F1* Score were used to determine the cotton detection and counting results. The calculation formula is as follows:

$$Precision = \frac{TP}{TP + FP}$$

$$Recall = \frac{TP}{TP + FN}$$

$$F1 = 2 \times \frac{Precision \times Recall}{Precision + recall}$$

TP is the true positives where the predicted result is cotton, the actual category is cotton, and the prediction is correct. TN is the true negatives where the prediction result is not cotton, a correct category is not cotton, and the prediction is correct. FP is the false positives where the proper category is not cotton, the prediction result is cotton, and the prediction is incorrect. FN is the false negatives where the correct category is cotton, the prediction result is not cotton, and the prediction is incorrect.

The coefficient of determination ($R^2$), root mean square error (*RMSE*), and relative root mean square error (*RRMSE*) were used as evaluation indexes to evaluate the performance of cotton seedling counting results.

$$R^2 = 1 - \frac{\sum_1^n (m_i - c_i)^2}{\sum_1^n \left(m_i - \bar{m}_i\right)^2}$$

$$RMSE = \sqrt{\frac{\sum_1^n (m_i - c_i)^2}{n}}$$

$$RRMSE = \sqrt{\frac{\sum_{i=1}^n (m_i - c_i)^2}{n}} \times \frac{1}{c_i}$$

where $m_i$, $\bar{m}_i$ and $c_i$ are, respectively, the number of artificially labeled cotton seedlings in the image $i$, the average number of artificially labeled cotton seedlings in the image $i$, and the predicted number of cotton seedlings. $n$ is the number of images tested.

### 3. Results

*3.1. Model Validation*

3.1.1. Model Training and Verification Test Results

As shown in Table 3, when deep learning cotton seedling detection was conducted based on YOLOv5, the performance at each time point was in the order of T3 > T6 > T4 > T2 > T5 > T1, with the best detection results at T3 and the worst at T1. When deep learning cotton seedling detection was conducted based on YOLOv7, the performance at each time point was in the order of T3 > T6 > T4 > T5 > T2 > T1. The detection results were the best at T3 and the worst at T1. When deep learning cotton seedling detection was conducted based on CenterNet, the performance at each time point was in the order of T6 > T3 > T5 > T4 > T2 > T1; the best results were detected at T6, and the worst results were detected at T1. In summary, when the Precision, Recall, and F1-Score were comprehensively evaluated, the performance of the three algorithms was in the order of YOLOv7 > YOLOv5 > CenterNet.

**Table 3.** Model training and validation test results.

| Model | Time | Training Set | | | Validation Set | | |
|---|---|---|---|---|---|---|---|
| | | Precision (%) | Recall (%) | F1-Score (%) | Precision (%) | Recall (%) | F1-Score (%) |
| YOLOv5 | T1 | 93.1 | 93.6 | 93.3 | 93.8 | 88.7 | 91.2 |
| | T2 | 95.9 | 95.2 | 95.5 | 96.6 | 96.0 | 96.3 |
| | T3 | 97.4 | 96.4 | 96.9 | 98.4 | 98.0 | 98.2 |
| | T4 | 96.1 | 95.7 | 95.9 | 96.5 | 95.4 | 95.9 |
| | T5 | 96.1 | 96.1 | 96.1 | 96.1 | 94.5 | 95.3 |
| | T6 | 97.7 | 95.7 | 96.7 | 97.2 | 94.6 | 95.9 |
| YOLOv7 | T1 | 92.3 | 94.4 | 93.3 | 94.3 | 92.0 | 93.2 |
| | T2 | 96.0 | 96.5 | 96.2 | 95.9 | 96.6 | 96.2 |
| | T3 | 97.3 | 98.2 | 97.7 | 98.1 | 98.6 | 98.3 |
| | T4 | 96.8 | 96.2 | 96.5 | 96.9 | 96.6 | 96.7 |
| | T5 | 95.8 | 96.4 | 96.1 | 97.3 | 96.5 | 96.9 |
| | T6 | 98.5 | 97.9 | 98.2 | 97.3 | 96.0 | 96.7 |
| CenterNet | T1 | 89.2 | 79.3 | 83.9 | 85.7 | 79.5 | 82.5 |
| | T2 | 93.3 | 82.8 | 87.7 | 92.9 | 82.6 | 87.4 |
| | T3 | 96.4 | 87.6 | 91.8 | 96.8 | 88.2 | 92.3 |
| | T4 | 95.1 | 83.1 | 88.7 | 94.8 | 83.8 | 89.0 |
| | T5 | 95.1 | 84.3 | 89.4 | 96.6 | 85.5 | 90.7 |
| | T6 | 98.2 | 93.2 | 95.6 | 97.9 | 93.4 | 95.6 |

3.1.2. Model Training and Verification of Counting Results

As shown in Table 4, based on the performance evaluation of YOLOv5 in cotton counting, the performance at each time point was in order of T2 > T4 > T3 > T5 > T1 > T6. The training and verification results at T2 were the best, while the training and verification results at T6 were the worst. Based on the performance evaluation of YOLOv7 in cotton counting, the performance of each time point was in the order of T2 > T3 > T5 > T4 > T1 > T6. The training and verification counting results were the best at T2 and the worst at T6. Based on the performance evaluation of CenterNet in cotton counting, the performance of CenterNet at each time was in the order of T2 > T3 > T4 > T5 > T6 > T1, with the best training and verification counting results at T2 and the worst at T1.

**Table 4.** Model training and verification counting results.

| Model | Time | Training Set | | | Validation Set | | |
|---|---|---|---|---|---|---|---|
| | | $R^2$ | RMSE | RRMSE (%) | $R^2$ | RMSE | RRMSE (%) |
| YOLOv5 | T1 | 0.85 | 9.36 | 8.40 | 0.64 | 12.48 | 11.46 |
| | T2 | 0.97 | 5.97 | 4.82 | 0.90 | 6.50 | 5.27 |
| | T3 | 0.91 | 9.53 | 4.55 | 0.93 | 9.97 | 4.73 |
| | T4 | 0.94 | 5.79 | 4.14 | 0.90 | 4.63 | 3.54 |
| | T5 | 0.94 | 7.89 | 5.69 | 0.87 | 5.41 | 3.84 |
| | T6 | 0.74 | 17.24 | 7.64 | 0.29 | 22.98 | 10.05 |
| YOLOv7 | T1 | 0.88 | 7.25 | 6.50 | 0.65 | 11.10 | 10.20 |
| | T2 | 0.97 | 11.48 | 9.27 | 0.91 | 5.21 | 4.22 |
| | T3 | 0.92 | 8.32 | 3.97 | 0.94 | 8.03 | 3.81 |
| | T4 | 0.92 | 5.82 | 4.16 | 0.90 | 4.32 | 3.31 |
| | T5 | 0.92 | 8.36 | 6.03 | 0.92 | 5.39 | 3.83 |
| | T6 | 0.78 | 9.17 | 4.06 | 0.57 | 11.61 | 5.08 |
| CenterNet | T1 | 0.88 | 10.55 | 9.46 | 0.68 | 12.75 | 11.71 |
| | T2 | 0.97 | 6.80 | 5.49 | 0.93 | 5.77 | 4.67 |
| | T3 | 0.95 | 6.66 | 3.18 | 0.94 | 8.55 | 4.06 |
| | T4 | 0.94 | 5.47 | 3.92 | 0.91 | 3.95 | 3.03 |
| | T5 | 0.92 | 6.87 | 4.96 | 0.97 | 3.36 | 2.39 |
| | T6 | 0.95 | 3.95 | 1.75 | 0.69 | 6.55 | 2.87 |

According to the performance of the above target detection algorithms, the counting accuracy of T2 was the highest, and the $R^2$ of T3, T4, and T5 are all higher than 0.90, and the performance was relatively stable. The summary shows that in the training and verification results based on YOLOv5, YOLOv7, and CenterNet, the counting results of YOLOv7 and CenterNet were better, while the counting results of YOLOv5 were slightly worse.

### 3.2. Model Test Results

Figure 6 show that cotton counting test results are obtained based on the above target detection algorithm. $R^2$, RMSE, and RRMSE indexes were used for model evaluation. When the cotton seedling counting test was conducted based on YOLOv5, the performance at each time point was successively T4 > T3 > T2 > T5 > T1 > T6. T4 had the best applicability of all the time points, while T6 had the worst result. When the test was conducted based on YOLOv7, the performance at each time point was successively T4 > T6 > T5 > T3 > T2 > T1. T4 had the best applicability in counting cotton seedlings, while T3 had the worst test result. In the test based on CenterNet, the performance at each time point was in the order of T2 > T5 > T4 > T1 > T6 > T3. T2 had the best applicability in the counting of cotton seedlings, and T3 had the worst test result.

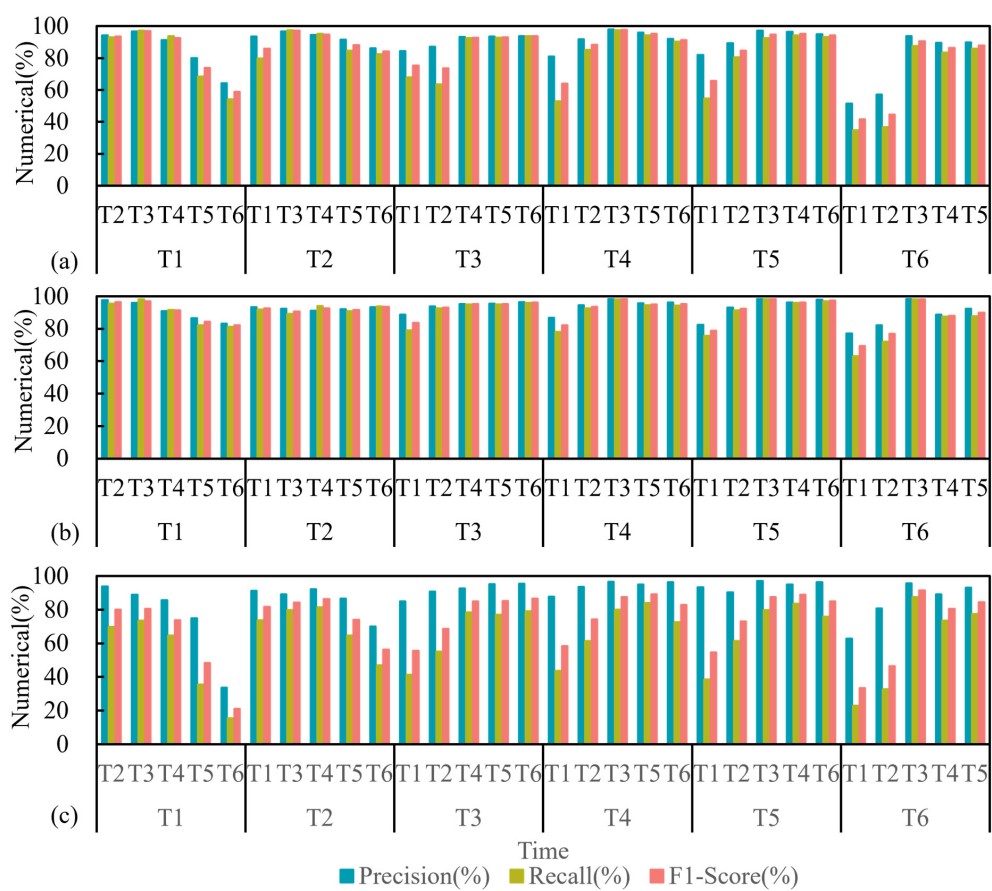

**Figure 6.** Model testing test results: (**a**) is YOLOv5, (**b**) is YOLOv7, and (**c**) is CenterNet.

As shown in Figure 7, the test results of cotton detection were obtained based on three target detection algorithms. The Precision, Recall, and F1-Score were used to evaluate the models. When tested based on YOLOv5, the performance at each time point was successively T4 > T5 > T3 > T2 > T1 > T6. T4 had the best applicability in cotton detection, while T6 had the worst test result. When the test was conducted based on YOLOv7, the performance of each time point was successively T4 > T3 > T5 > T6 > T2 > T1. T4 had the best applicability in cotton detection, while T6 had the worst test result. In the test

based on CenterNet, the performance at each time point was in the order of T4 > T5 > T3 > T2 > T6 > T1. T4 had the best applicability in cotton detection, while T1 had the worst test result.

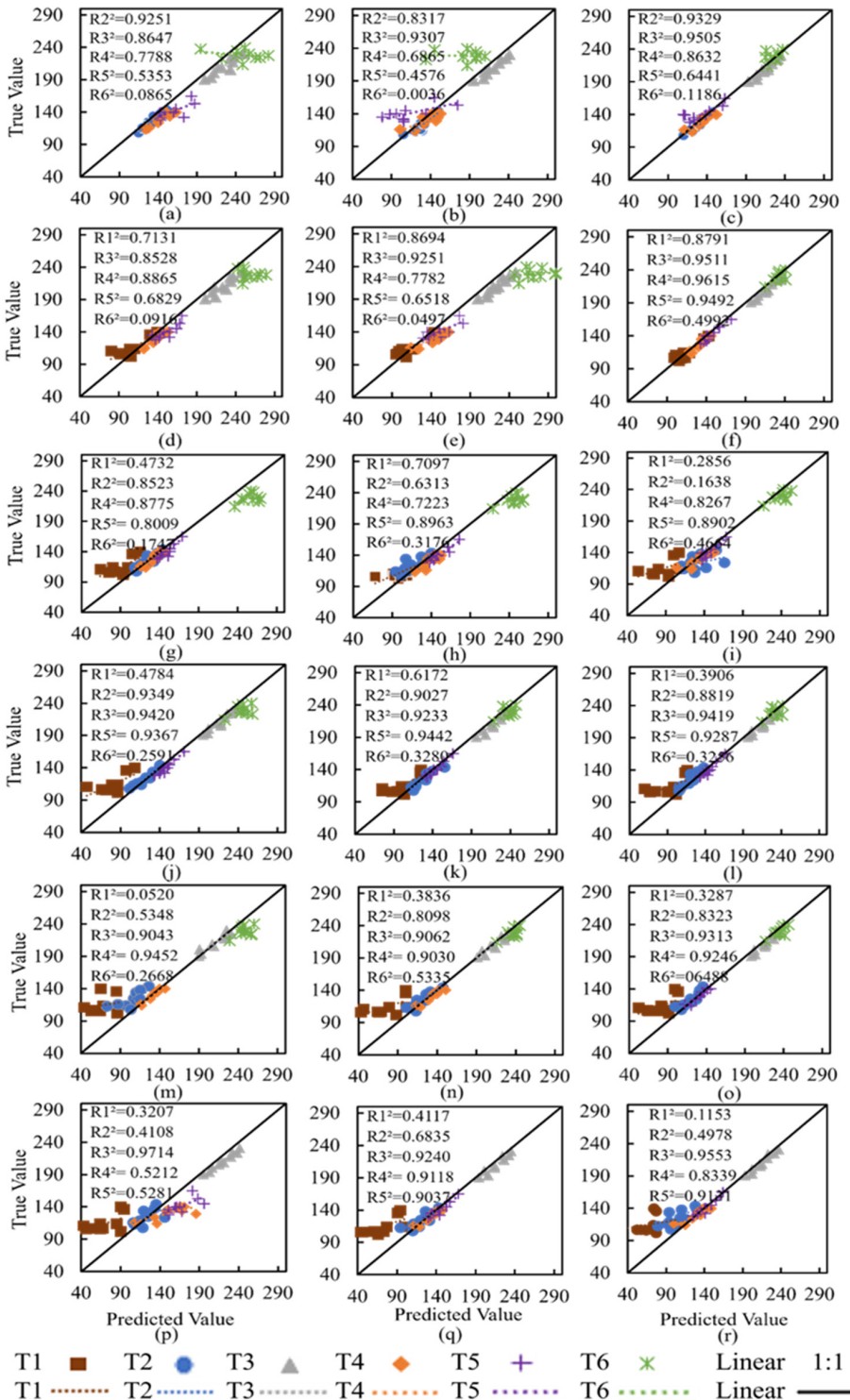

**Figure 7.** The results of YOLOv5, YOLOv7, and CenterNet counting tests at T1–T6: Each row from top to bottom represents the time points T1–T6, and each column from left to right represents YOLOv5, YOLOv7, and CenterNet. (**a,d,g,j,m,p**) represent the test results of YOLOv5 from T1–T6; (**b,e,h,k,n,q**) represent the test results of YOLOv7 during T1–T6, and (**c,f,i,l,o,r**) represent the test results of CenterNet during T1–T6.

Figure 8 shows an example of cotton seedling detection and counting at T4. The three algorithms all overestimated the number of cotton seedlings, and visually interpreted one hundred and thirty-seven cotton seedlings. The YOLOv5 algorithm detected one hundred and forty-one cotton seedlings, and four were false positives. The YOLOv7 algorithm detected one hundred and forty cotton plants with three false positives. CenterNet algorithm detected one hundred and forty-four seedlings, with seven false positives.

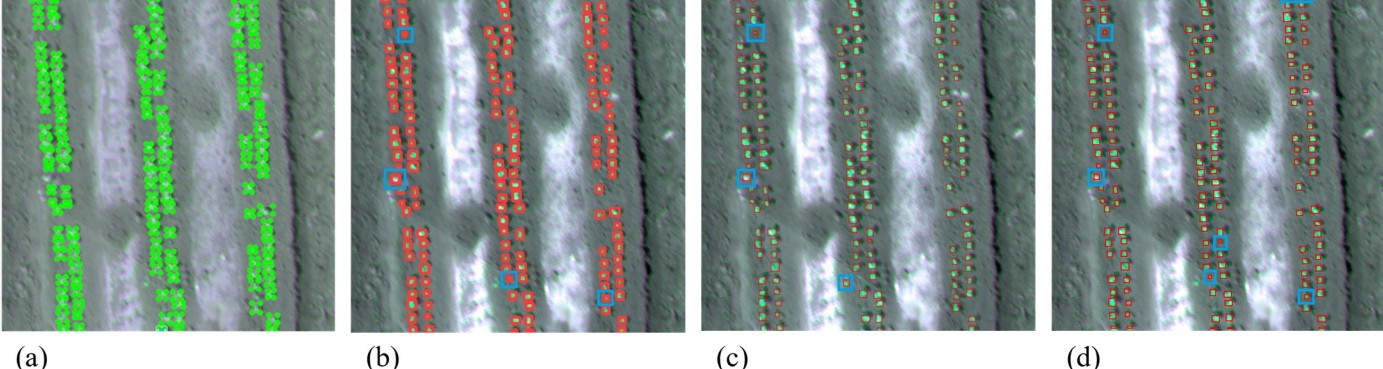

(a)　　　　　　　　(b)　　　　　　　　(c)　　　　　　　　(d)

**Figure 8.** An example of the results from the three algorithms for counting cotton seedlings in the T4 dataset: (**a**) is the manually labeled cotton seedling image, (**b–d**) represent the counting results using the YOLOv5, YOLOv7 and CenterNet algorithms, respectively; yellow boxes indicate missed detections, blue boxes indicate false positives.

Figure 9 shows the test results at different time points compared to T4 based on the YOLOv7 algorithm. A total of one hundred and fourteen cotton seedlings were detected in one image at T1. There were one hundred and fourteen cotton seedlings in an image at T2. The algorithm overestimated the number of seedlings and detected one hundred and eighteen cotton seedlings, with six false positives and two false negatives. A total of two hundred and one cotton seedlings were detected in an image during the T3 period. The algorithm overestimated the number of seedlings and detected two hundred and five cotton seedlings, with five false positives and one false negatives. There were one hundred and thirty-nine cotton seedlings in an image during the T5 period. The algorithm overestimated the number of seedlings and detected one hundred and forty-two cotton seedlings, with five false positives and two false negatives. There were two hundred and twenty-five cotton seedlings in an image during the T6 period. The algorithm overestimated the number of seedlings and detected 235 cotton seedlings, with 11 false positives and 1 false negative. The number of specific cotton seedlings detected and the number of false positives in the training and validation set of the T4 dataset are shown in Table 5.

**Table 5.** Number of specific cotton seedlings and false positives detected in the training and validation sets of the T4 dataset.

| Model | Training Set | | | | Validation Set | | | |
|---|---|---|---|---|---|---|---|---|
| | True Value | Predicted Value | False Positive | False Negative | True Value | Predicted Value | False Positive | False Negative |
| YOLOv5 | 4192 | 4334 | 150 | 8 | 1305 | 1339 | 37 | 3 |
| YOLOv7 | 4192 | 4324 | 135 | 3 | 1305 | 1334 | 31 | 2 |
| CenterNet | 4192 | 4326 | 140 | 6 | 1305 | 1333 | 31 | 3 |

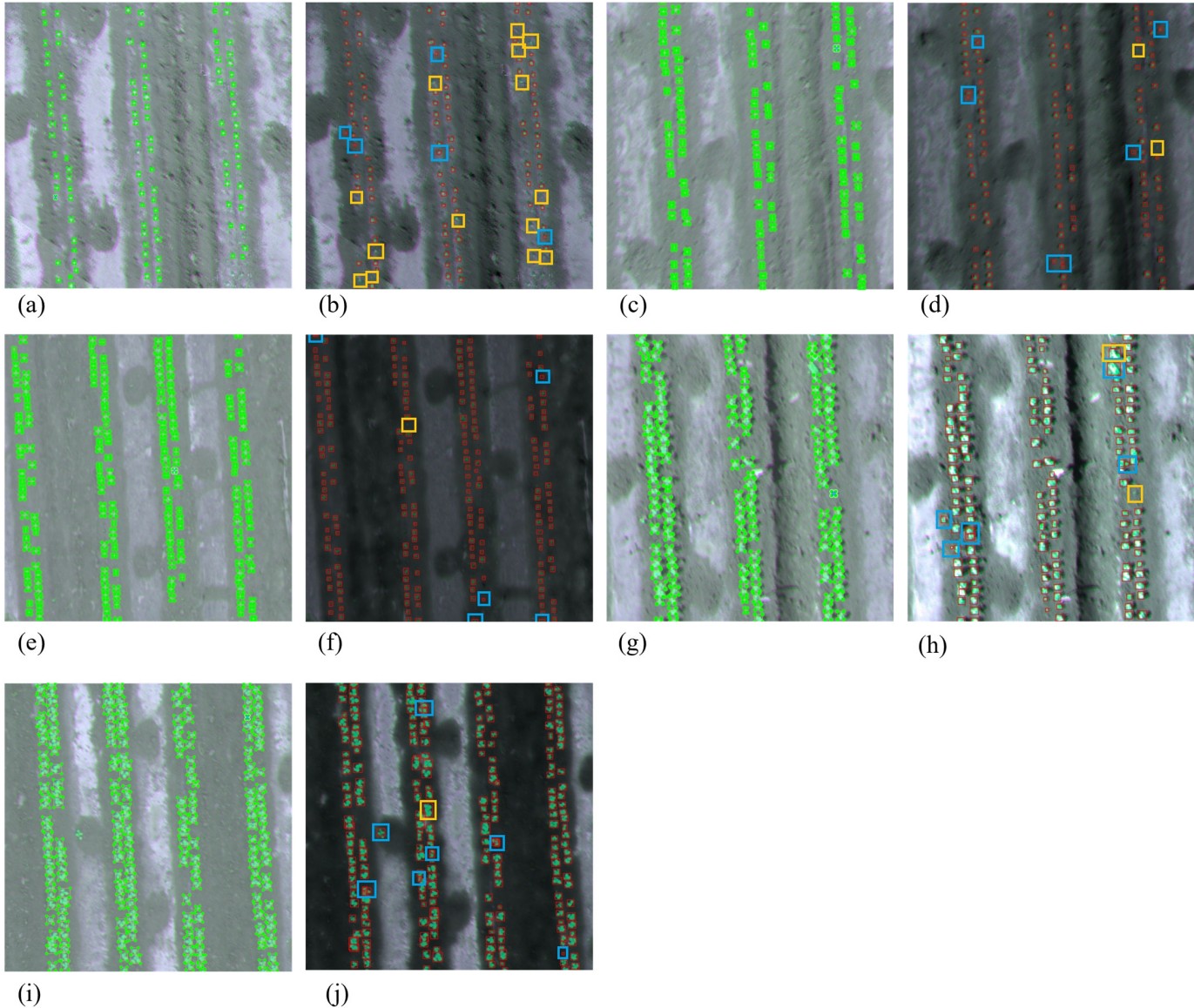

**Figure 9.** The T4 dataset was tested using YOLOv7, as well as the counts of the model in the other five datasets: (**a,c,e,g,i**) indicate manually labeled images in the T1, T2, T3, T5, and T6 datasets, respectively. (**b,d,f,h,j**) correspond to the count results of YOLOv7. The yellow box in the figure indicates a missed detection, and the blue box indicates a false detection. (**f,j**) indicate T3 and T6, which were cloudy at the time of image acquisition.

## 4. Discussion

As cotton seedlings gradually grew from small to large, the number of cotton leaves gradually increased. There were significant differences between the detection and counting models at different time points. Therefore, the image acquisition time point is particularly critical in cotton detection and counting, which determines cotton size. If the images are acquired too early, the seedlings are too small to be detected and counted, e.g., seedling detection and counting at T1 in this study. If the images are acquired too late, the cotton seedlings reach the four-leaf stage, and the leaves have spread out, with crossing and overlap occurring between leaves, resulting in underestimation of the cotton population, e.g., seedling detection and counting at T6 in this study, which are also the two significant challenges encountered so far [51]. Due to the short window of cotton seedlings, if the best time to replenish them is missed, it will affect the cotton yield at the later harvest. To overcome these challenges, we should mainly improve

the algorithms, such as YOLOv5 and YOLOv7, which are more effective in small target detection, and CenterNet, which is more effective in detecting seedlings when they are larger, in order to combine the advantages of each algorithm to improve and solve the significant challenges caused by smaller seedlings and overlapping leaves. Because the growth rate of cotton seedlings depends on different environmental factors, such as soil type, temperature, water, and fertilizer, and weather conditions, in this study, the image collection time was from ten days after seeding to twenty-seven days after seeding, with a total of six time points for collection, which is the reference value for determining the optimal detection and counting time.

The resolution of multispectral images is mainly determined by the UAV's camera or sensor and the UAV's flying altitude. Better cameras and sensors represent a high cost. The standard minimum flying altitude is ten meters, and it is recommended to fly at most ten meters. This study used a multispectral camera to obtain multispectral images of the red, green, and near-infrared bands with sound detection and counting accuracy [52]. A multispectral sensor collected images for cotton plant detection and counting, and good results were also obtained. Compared with RGB cameras, we obtained more information about the cotton, and the presence of cotton in the image was more prominent [53,54].

The vast majority of cotton grown in Xinjiang province is sprayed with herbicide before sowing, a process that minimizes the impact of weeds. Then, machines are used to lay drip irrigation belts, planters, and plastic film. Placing the plastic film keeps weeds from growing between the cotton seedlings, ensuring that weeds do not interfere with target detection and competition for cotton nutrients. In order to prevent the plastic film from blowing away in strong winds, the plastic film is covered with a layer of soil at a specific interval. However, although this layer of soil solves the risk of the film blowing away, it also increases the difficulty of the growth of cotton seedlings [55]. About ten days after sowing, seedlings should be released. In order to prevent seedlings from being burned to death at high temperatures, seedlings should be able to grow normally from between the plastic film. Due to the small size of the cotton seedlings, some of them were not taller than the plastic film, such as the T1 period in this study, so the counting accuracy was complicated.

In this paper, cotton leaves included two–four leaves from ten days to twenty-seven days after sowing, and the morphological diversity of cotton was relatively affluent. According to the analysis of the detection and counting effects from training verification and test results, model training speed, and weight file size, the overall performance of the YOLOv7 algorithm is promising. YOLOv7 extends the E-ELAN network based on YOLOv5, and extends, disorders, and combines Cardinality [56], which can improve the network's learning ability without destroying the original gradient path. In E-ELAN's network architecture, the transition layer architecture was not changed, only the architecture in the fast computing group. The scaling method of this model is a compound model scaling method. When scaling a depth factor with fast calculation, the change of the output channel of the block should also be calculated, and the width factor scaling of the transition layer should be carried out with the exact change amount. The convolution reparameterization is introduced and improved, and the auxiliary training module is introduced to generate the level of labels from coarse to fine. The detection time of each image was 16.71 ms, and when the training step is 1000, the weight file size was 71.3 MB. YOLOv5 took 131.7 ms to detect each image, and the weight file size was 13.7 MB when the training step size was 300. The YOLOv5 algorithm is a single-stage target detection algorithm. Images at the input end are spliced using random scaling, clipping, and arrangement. This data enhancement method combines several images into one, which can not only enrich the dataset but also improves the training speed and, most importantly, reduces the memory requirements of the model. A method of adaptive anchor frame calculation and adaptive picture scaling was proposed. In terms of the benchmark network, YOLOv5 uses the CSPDarknet53 structure and the

Focus structure as the benchmark network. This module can extract some common feature representations. Although the training speed and detection accuracy could be better than YOLOv7, the small model is suitable for embedding in the mobile end [57], which provides the possibility for cotton field detection and counting. The main idea of the CenterNet algorithm is to treat the target as a critical point when creating the model, i.e., the center point of the target boundary frame [58]. The algorithm finds the central point by evaluating the critical point of the target and makes regression predictions on other attributes of the target. This algorithm had a better detection and counting effect on slightly overlapping cotton seedlings than the other two algorithms.

Weather conditions have a significant influence on field image acquisition by UAVs. In this study, the time of image collection was from noon to 5 PM. Under the background of long sunshine in Xinjiang, the light intensity in this period was the highest. Among them, the weather was cloudy when the images were collected at T3 and T6, and the soil background and cotton seedlings were dark in the images. However, the detection and counting of cotton seedlings using the three algorithms had better results. In this study, multispectral images on cloudy days were collected as a reference, and the results of cotton seedling detection and counting were good, similar to previous studies [9,10,52,55]. The results showed that in this study, six time periods between ten days and twenty-seven days after the sowing of cotton were collected for plant detection and counting, and the best period for detection and counting was determined. After determining the optimal detection time, a generalized model with high generalization ability was constructed at the cotton seedling stage. The growth rate of cotton at the seedling stage was very uneven, which changed considerably within two–three days. Therefore, the time division could be more detailed based on this study. In future studies, the application of different crops with large planting areas in Xinjiang, such as maize, wheat, and sugar beet, and different imaging techniques should be considered to validate the robustness of the model. In addition, CenterNet can be improved by integrating other detection algorithms that have a better counting effect for cotton with an overlapping phenomenon to improve the overall detection and counting effect.

With the continuous development of deep learning, AI algorithm-driven software products such as target detection-type applications are increasingly affecting our lives. The technology is developing in a longitudinal direction, such as the continuous iteration of autonomous driving. Applying formal methods for analyzing and validating AI software in future research should focus on data reliability issues: (1) whether the data set is evenly distributed; (2) the size of the dataset used to train the model; and (3) whether the samples are appropriately labelled. It is hoped that formal methods will be applied to solve the problems related to security and validation in this area of AI [59,60].

## 5. Conclusions

In this study, cotton images of three bands were collected by a Sprite 4 multispectral version UAV, and the acquired multispectral images were processed. Deep learning was conducted based on YOLOv5, YOLOv7, and CenterNet target-detection algorithms to obtain cotton seedling detection and counting results. For cotton seedling detection, the algorithm performance in the process of model construction was in the order of YOLOv7 > YOLOv5 > CenterNet, and the algorithm performance at each time point was in the order of T3 > T6 > T4 > T5 > T2 > T1. During the counting of cotton seedlings, the algorithm performance in the process of model construction was in the order of CenterNet > YOLOv7 > YOLOv5, and in the order of T2 > T3 > T4 > T5 > T6 > T1. When the data collected at different times were tested, the all-around performance of YOLOv7 in detecting and counting cotton seedlings in the T4 period was better. In summary, (1) six images at different time points were used for model verification, and YOLOv7 was the optimal algorithm; and (2) based on these three target detection algorithms, T4 was the most suitable time point for seedling detection and counting in the cotton seedling stage. Therefore, the multispectral images of UAV obtained at the T4 time point of the cotton seedling stage and

the YOLOv7 algorithm were used to realize rapid and accurate detection and counting of cotton seedlings in the whole seedling stage. This approach can provide technical support for checking seedling and repairing seedlings in time and provide the foundation for cotton production with high efficiency, high quality, and high yield.

YOLOv5, YOLOv7, and CenterNet algorithms can accurately detect and count cotton seedlings, and the combination of target detection algorithms and UAV remote sensing for counting has a broad application prospect. However, there are significant challenges for counting under the conditions of small seedling size and considerable overlap. In future research, we need to further investigate the effects of light conditions, image resolution, and complex backgrounds on cotton seedling detection accuracy. Evaluating the optimal flight height and image resolution is another worthwhile direction.

**Author Contributions:** Conceptualization, P.G., X.L. and Y.F.; methodology, Y.F. and W.C.; software, Y.F., W.C. and Y.M.; validation, Y.F. and W.C.; formal analysis, Y.F.; investigation, Y.F., W.C. and Y.M.; resources, P.G., X.L. and Z.Z.; data curation, Y.F.; writing—original draft preparation, Y.F.; writing—review and editing, P.G. and X.L.; visualization, Y.F.; supervision, P.G. and X.L.; project administration, P.G. and X.L.; funding acquisition, P.G. and X.L. All authors have read and agreed to the published version of the manuscript.

**Funding:** This study was supported by the National Natural Science Foundation of China (Grant No. 62265015).

**Data Availability Statement:** Not applicable.

**Conflicts of Interest:** The authors declare that the research was conducted in the absence of any commercial or financial relationships that could be construed as a potential conflict of interest.

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
