# Peer review of "Cotton Seedling Detection and Counting Based on UAV Multispectral Images and Deep Learning Methods"

_remotesensing, doi:10.3390/rs15102680_

Round 1
Reviewer 1 Report
This manuscript discussed cotton seedling detection and counting using UAV-based spectral images and deep learning. It is a good research with promising results. However, a lot of similar studies were published. Can the authors talk about your novelty? In Table 2, the testing set have more samples than training set, why? Besides, the authors said it is 3:1:1, but it is not 3:1:1 in Table 1. In Figure 1, why not collected the images from the vertical review? Also, the quality of some images is poor, such as Figure 5. Some wiring issues, such as R2 in line 22.
Minor editing of English language required
Author Response
Response to Reviewer 1 Comments
General comment: This manuscript discussed cotton seedling detection and counting using UAV-based spectral images and deep learning. It is a good research with promising results.
Response: We thank the reviewer very much for these positive and encouraging comments. In particular, we really appreciate the referee for her/his suggestions. Please see our responses to the detailed comments below. For convenience, all changes in the revised manuscript were highlighted using the MS Word“Track Changes” function.
Point 1: Can the authors talk about your novelty?
Response 1: We apologize for this error, which we did not clearly express. In this study, we present three innovations, (1) Selecting the optimal algorithm using the performance of YOLOv5, YOLOv7, and CenterNet in the six times of collected UAV multispectral images. (2) Selecting the optimal cotton seedling detection and counting time-based on the selected optimal algorithm. (3) Final selection of a cotton seedling detection and counting model applicable to the entire seedling period. We made changes on page 3, lines 124-128.
Point 2: In Table 2, the testing set have more samples than training set, why?
Response 2: We are very sorry for this error. We applied validation sets from other times as the test dataset for a specific time when we did the test. However, we tested the models individually instead of testing the models with the validation sets several other times. For example, the test result for T1 tests the detection and counting model constructed from the T1 dataset against the validation sets of the T2, T3, T4, T5, and T6 datasets, respectively. Initially, we added up the number of cotton seedlings from all test sets, which is why there are more cotton seedlings in the test set than in the training set. We have modified the presentation problem on page 5, lines 185-195, and Table 2.
Table 2. The number of cotton seedling marks in training, verification and testing images at different periods.
Time |
Train set |
Val(Test) set |
Total |
T1 |
3234 |
1143 |
4377 |
T2 |
3715 |
1235 |
4950 |
T3 |
6284 |
2108 |
8392 |
T4 |
4192 |
1305 |
5497 |
T5 |
4160 |
1408 |
5568 |
T6 |
6770 |
2286 |
9056 |
Total |
28355 |
9485 |
37840 |
Point 3: Besides, the authors said it is 3:1:1, but it is not 3:1:1 in Table 1.
Response 3: Again, we apologize for this error, and we split the training and validation of the dataset into 3:1 at each time, with the test set at one time being the validation set at other times. As answered in Response 2, we have modified Table 2.
Table 2. The number of cotton seedling marks in training, verification and testing images at different periods.
Time |
Train set |
Val(Test) set |
Total |
T1 |
3234 |
1143 |
4377 |
T2 |
3715 |
1235 |
4950 |
T3 |
6284 |
2108 |
8392 |
T4 |
4192 |
1305 |
5497 |
T5 |
4160 |
1408 |
5568 |
T6 |
6770 |
2286 |
9056 |
Total |
28355 |
9485 |
37840 |
Point 4: In Figure 1, why not collected the images from the vertical review?
Response 4: We apologize for this error. Originally, d in Figure 1 was the original image taken by the drone. However, the test area was not directly south and north, and the course direction set in the drone route planning, considering the influence of the sun angle, resulted in a skewed image. After you raised this issue, we also realized that this would not show the content clearly, and we changed Figure 1d on page 4, line 146.
Figure 1. Study area overview : a is the map of Xinjiang, b is the map of Shihezi, c is the test site, and d is the multispectral images collected by the UAV.
Point 5: Also, the quality of some images is poor, such as Figure 5. Some wiring issues, such as R2 in line 22.
Response 5: We apologize for the error and appreciate the reviewers' careful manuscript reading. We have rechecked the error in the article and corrected it by changing "R2" to "R2" on page 1, line 24. We have also redrawn the connection line in Figure 5 and corrected it on page 7, line 280.
Figure 5. CenterNet network structure.

Reviewer 2 Report
Summary/Contribution: This work employs three deep learning algorithms, YOLOv5, YOLOv7, and CenterNet, to detect and count seedling cotton in multispectral UAV photos during six different stages of the cotton seedling period. The work seeks to develop a model for the complete cotton seedling stage and to detect and count seedlings quickly and accurately. YOLOv7 performs higher overall detection and counting, while the T4 dataset performs better in each test set. The UAV multispectral photos and YOLOv7 algorithm produced the anticipated results 23 days after cotton seeding (T4).
Comments/Suggestions:A. Some strong points of this work are:
1. Practical significance: The study addresses the practical need for timely seedling inspection and replenishment at the seedling stage, which is crucial for cotton's late production management and yield formation.
2. Innovative approach: The study proposes the use of multispectral images collected by UAVs at different times during the cotton seedling period and the application of deep learning algorithms, YOLOv5, YOLOv7, and CenterNet, for cotton seedling detection and counting.
B. Some limitations of this work are:
1. Limited validation: The study's findings are based on the performance of the deep learning algorithms on the dataset collected in Xinjiang, and the generalizability of the results to other regions or crops is unclear.
2. Lack of cost-effectiveness analysis: The study did not consider the cost-effectiveness of the proposed approach compared to traditional methods of seedling inspection and replenishment.
3. Limited scope: The study focused only on cotton seedling detection and counting using multispectral images collected by UAVs, and the results may not be generalizable to other crops or imaging techniques.
4. Lack of comparison with traditional methods: The study did not compare the performance of the deep learning algorithms with traditional methods used for cotton seedling detection and counting.
C. What is the significance of seedling emergence rate, and how does it affect crop yield? Were there any challenges or limitations encountered during the investigation, and how did the researchers address them?
I. Is it possible to evaluate the effects of light conditions, image resolution, and complex background on the detection accuracy of cotton seedlings, and determine the optimal flight altitude and image resolution for improving the accuracy of cotton seedling detection and counting using UAV remote sensing?
Author Response
Response to Reviewer 2 Comments
General comment: This work employs three deep learning algorithms, YOLOv5, YOLOv7, and CenterNet, to detect and count seedling cotton in multispectral UAV photos during six different stages of the cotton seedling period. The work seeks to develop a model for the complete cotton seedling stage and to detect and count seedlings quickly and accurately. YOLOv7 performs higher overall detection and counting, while the T4 dataset performs better in each test set. The UAV multispectral photos and YOLOv7 algorithm produced the anticipated results 23 days after cotton seeding (T4).
Response: We are very grateful for the reviewer's comments, and we will revise them carefully according to the comments. Please see our responses to the detailed comments below. For convenience, all changes in the revised manuscript were highlighted using the MS Word “Track Changes” function.
Point 1 A. Some strong points of this work are:
- Practical significance: The study addresses the practical need for timely seedling inspection and replenishment at the seedling stage, which is crucial for cotton's late production management and yield formation.
- Innovative approach: The study proposes the use of multispectral images collected by UAVs at different times during the cotton seedling period and the application of deep learning algorithms, YOLOv5, YOLOv7, and CenterNet, for cotton seedling detection and counting.
Response 1: We are very grateful to the reviewers for these positive and encouraging comments. We are incredibly grateful to the reviewers for carefully reviewing the manuscript.
Point 2 B. Some limitations of this work are:
- Limited validation: The study's findings are based on the performance of the deep learning algorithms on the dataset collected in Xinjiang, and the generalizability of the results to other regions or crops is unclear.
- Lack of cost-effectiveness analysis: The study did not consider the cost-effectiveness of the proposed approach compared to traditional methods of seedling inspection and replenishment.
- Limited scope: The study focused only on cotton seedling detection and counting using multispectral images collected by UAVs, and the results may not be generalizable to other crops or imaging techniques.
- Lack of comparison with traditional methods: The study did not compare the performance of the deep learning algorithms with traditional methods used for cotton seedling detection and counting.
Response 2: 1. Thank you very much for pointing out the shortcomings in this paper with limited validation. Xinjiang is China's most significant and crucial cotton production area, accounting for 82.76% of the national cotton cultivation area and 89.5% of the total national production in 2021. Therefore, we would like to include cotton or other crops from other regions in Xinjiang in our subsequent study to validate the robustness of the model. We made changes on page 21, lines 496-498.
- Thank you very much for pointing out the shortcomings in this paper and the lack of cost-benefit analysis. At present, RGB images are primarily applied in the research on target detection and counting, and in the pursuit of high accuracy, the imaging requirements of RGB cameras are getting higher and higher; of course, this pursuit also means an increase in cost, and the resolution of images acquired by RGB sensors is lower than that of spectral sensors at the same image size, so multispectral images are applied in this study for cotton seedling detection and counting research. We have made changes on page 2, lines 56-58.
- Thank you for pointing out the deficiencies in the paper, which are limited in scope. Currently, one crop, cotton, is mainly considered in this study, and in future research, to improve the robust performance of the model, it is crucial to further apply it in other crops and in imaging techniques to make the model generalization performance stronger. Changes were made on page 21, lines 496-498.
- Thank you for pointing out the shortcomings of the paper and the lack of comparison with traditional methods. When reviewing the literature to determine the direction of research, traditional methods for seedling detection and counting have limitations: traditional machine learning algorithms such as random forest (RF) and support vector machines (SVM) can perform fast classification of images. Support vector machine models can be used to classify spectral and texture features and estimate seedlings' number and density. In contrast, for supervised classification methods, manually selecting a sufficient number of representative samples from many images is tedious and time-consuming, and the manual selection can also reduce classification accuracy. Therefore, this study considers applying deep learning methods for cotton seedling detection and counting to significantly reduce the errors caused by human labour and human selection. We made changes on page 2, lines 79-86.
Point 3 C. What is the significance of seedling emergence rate, and how does it affect crop yield? Were there any challenges or limitations encountered during the investigation, and how did the researchers address them?
Response 3: We apologize for this error. The statistical significance of seedling emergence is to count the number of cotton seedlings in an area as a percentage of the theoretical number of seedlings preserved and to provide a theoretical basis for timely replenishment.
An accurate estimate of the number of plants in a field crop is essential for predicting yield. We have modified lines 34-35 on page 1 and have cited relevant literature.
The challenges encountered during manual surveys are mainly high cost, high workload, poor timeliness, inaccurate counts, and poor data representativeness, which could be more conducive to timely replanting. We made changes on page 1, lines 36-39.
Ultimately, we wanted to solve the challenges encountered in the manual survey by acquiring multispectral images of the UAV at different times for seedling detection and counting model construction. We made changes on page 3, lines 121-122.
- Kitano, B. T., Mendes, C. C. T., Geus, A. R., Oliveira, H. C., and Souza, J. R., 2019. Corn plant counting using deep learning and UAV images. IEEE Geosci. Remote Sens. Lett., 1–5. https://doi.org/10.1109/lgrs.2019.2930549
- Oliveira, H. C., Guizilini, V. C., Nunes, I. P., and Souza, J. R., 2018. Failure detection in row crops from UAV images using morphological operators. IEEE Geosci. Remote Sens. Lett.,15(7), 991–995. https://doi.org/10.1109/LGRS.2018.2819944
Point 4 D. What are the main differences between CenterNet and traditional target detection algorithms that frame the target in a rectangular box, and how does CenterNet construct its model for target detection and recognition? Additionally, what were the specific parameters used for the image input size, training weight, optimizer, batch size, and epoch during the implementation of CenterNet in this study?
Response 4: We apologize for this error, and thank you very much for pointing out the problem. The main difference between CenterNet and traditional target detection algorithms (framing the target in a rectangular box) is that CenterNet is an anchorless target detection algorithm that improves on the Cor-nerNet algorithm. The algorithm removes inefficient complex Anchors, performs filtering operations directly on the heat map, removes time-consuming NMS post-processing operations, transforms the target detection problem into a critical point detection problem, and further improves the performance of the detection algorithm. In contrast to CenterNet, most target detection and recognition usually frames the target in a rectangular box. The horizontal and vertical axes of the rectangular box are parallel to the horizontal and vertical axes of the image. These detectors identify potential target locations and then classify them, which wastes time and requires additional post-processing. centerNet builds its target detection and recognition model so that centerNet does not require manual thresholding of foreground or background by assigning anchors based on location rather than dense coverage on the feature map. Each object in it has only one positive sample of anchor, so there is no need for non-maximal value suppression processing, i.e., no post-processing methods are needed; only local peaks in the critical point heatmap need to be extracted. A larger output resolution is used, which eliminates the need for multiple anchors [48]. The primary process of building its target detection and recognition model is as follows, the image is fed into a fully convolutional network used to generate the heatmap, the peak position of the heatmap is the centre of the target, and the image features at each peak position are used to predict the width and height of the target bounding box. Its network structure diagram is shown in Figure 5. The image input size is 640×640, the initial training weights file is centernet_resnet50_voc, the optimizer is chosen as adam, the batch size is 4, and the epoch is 900. we made changes on page 9 and page 10, lines 245-265.
Point 5 E. The authors should include a paragraph on how formal methods might be used to verify AI-based solutions, notably data collection and processing. For this purpose, the following references may be included: a. https://ieeexplore.ieee.org/document/9842406 b. https://dl.acm.org/doi/abs/10.1145/3503914
Response 5: Thank you very much for the advice you provided. With the continuous development of deep learning, AI algorithm-driven software products such as target detection-type applications are increasingly affecting our lives. The technology is developing in a longitudinal direction, such as the continuous iteration of autonomous driving technology. Applying formal methods for analysing and validating AI software in future research focuses on data reliability issues, (1) whether the data set is evenly distributed. (2) The size of the dataset used to train the model. (3) Whether the samples are appropriately labelled. It is desired to apply formal methods to address issues related to security and validation in this area of artificial intelligence. We have revised it on pages 21, 504-511.
- Krichen, M.; Mihoub, A.; Alzahrani, M.Y.; Adoni, W.Y.H.; Nahhal, T. Are Formal Methods Applicable To Ma-chine Learning And Artificial Intelligence? In Proceedings of the 2022 2nd International Conference of Smart Systems and Emerging Technologies (SMARTTECH); IEEE: Riyadh, Saudi Arabia, May 2022; pp. 48–53.
- Seshia, S.A.; Sadigh, D.; Sastry, S.S. Toward Verified Artificial Intelligence. Commun. ACM 2022, 65, 46–55, doi:10.1145/3503914.
Point 6 F. What is the significance of the results of the three algorithms' performance in detecting and counting cotton seedlings during T4, as shown in Figure 8? Additionally, what were the specific numbers of cotton seedlings detected and the numbers of false positives for the YOLOv5, YOLOv7, and CenterNet algorithms during T4?
Response 6: We are very sorry for this error. The significance of the detection and counting results of the three algorithms in the T4 dataset is mainly the following: (1) Firstly, we used the three algorithms to construct models in the six datasets T1-T6 and selected the best-performing dataset among the three algorithms. (2) Secondly, according to the selected data set, we can roughly determine an optimal time for cotton seedling detection and counting. (3) Finally, a generalized detection model with high generalization ability is constructed at the cotton seedling stage. We specifically modify this on page 21, lines 492-493.
The number of cotton seedlings detected explicitly by the three algorithms at T4 time with the number of false positives is shown in Table 5 and modified in rows 400-401 on page 19.
Table 5. Number of specific cotton seedlings and false positives detected in the training and validation set of the T4 dataset.
Model |
Train set |
Val set |
|||||||
True value |
Predicted value |
False positive |
False negative |
True value |
Predicted value |
False positive |
False negative |
||
YOLOv5 |
4192 |
4334 |
150 |
8 |
1305 |
1339 |
37 |
3 |
|
YOLOv7 |
4192 |
4324 |
135 |
3 |
1305 |
1334 |
31 |
2 |
|
CenterNet |
4192 |
4326 |
140 |
6 |
1305 |
1333 |
31 |
3 |
|
Point 7 G. What is the significance of image acquisition time in cotton seedling detection and counting, and how does it affect the accuracy of the detection and counting models? Additionally, how did the researchers determine the image collection time in this study, and what factors did they consider in making this determination?
Response 7: We are sorry for this error. The significance of the timing of the image collection is mainly due to the short window of cotton seedlings, which will affect the yield of cotton in later harvests if the best time to replenish seedlings is missed. Changes were made in the discussion section on page 21, lines 420-421.
Two main factors of image acquisition time affect the accuracy of the detection and counting model: If the image acquisition time is too early, cotton seedlings are small and difficult to detect and count, e.g., seedling detection and counting performed at T1 time in this study. If the images are acquired too late, cotton seedlings reach the four-leaf stage, and the leaves have spread out, with crossover and overlap between leaves, leading to underestimation of cotton numbers, e.g., seedling detection and counting performed at T6 time in this study, which is also the two main challenges encountered so far. Changes were made on page 19 and page 20, lines 407-413.
Our primary basis for determining the timing of image acquisition is as described below. The cotton seedling stage is roughly 30-40 days, and it takes roughly ten days from sowing until most cotton seedlings grow out of the plastic film. Images are acquired every three days, starting on day ten after sowing, throughout the seedling stage. The main factor is whether the image acquisition time can cover the whole seedling period.
Point 8 H. The authors are invited to explain the significant challenges faced by the YOLOv5, YOLOv7, and CenterNet algorithms in accurately detecting and counting cotton seedlings, and suggest ways to address them.
Response 8: We apologize for this error. The three algorithm faces two main challenges for cotton seedling detection and counting. (1) If the images are acquired too early, the seedlings are tiny, and some still need to be as high as the film, making detecting and counting cotton seedlings difficult. (2) If the images are acquired late, the cotton seedlings are left with four to five or six leaves, and the leaves have already spread out and are blocking each other, which also makes it more difficult to detect and count the cotton seedlings, and we have made changes on page 19 and page 20, lines 407-413.
Facing these challenges, our main idea is to improve the algorithms, such as YOLOv5 and YOLOv7, which are more effective in small target detection, and CenterNet, which is more effective in detecting larger seedlings, to consider combining the advantages of each algorithm to improve and solve the significant challenges caused by smaller seedlings and overlapping leaves. We have made changes on page 20, lines 421-426.
Point 9 I. Is it possible to evaluate the effects of light conditions, image resolution, and complex background on the detection accuracy of cotton seedlings, and determine the optimal flight altitude and image resolution for improving the accuracy of cotton seedling detection and counting using UAV remote sensing?
Response 9: We apologize for this error, which we did not express clearly in the text. In future research, we need to investigate further the effects of light conditions, image resolution and complex background on cotton seedling detection accuracy. Evaluating the optimal flight height and image resolution is a worthwhile direction to investigate. We have made changes on page 22, lines 535-538.

Round 2
Reviewer 1 Report
The authors have answered all my questions. It can be accepted now.
No questions.
Reviewer 2 Report
The authors considered all my comments and suggestions. Good luck.
May be improved.